# Spinal Intradural Hematoma after Spinal Anesthesia in a Young Male Patient: Case Report and Review of the Literature

**DOI:** 10.3390/ijerph19084845

**Published:** 2022-04-16

**Authors:** Jae Young Ji, Jae Min Ahn, Jin Hun Chung, Nan Seol Kim, Yong Han Seo, Ho Soon Jung, Hea Rim Chun, Woo Jong Kim, Chan Ho Park, Jeong Soo Choi, Hyun Chul Jung, Jin Soo Park

**Affiliations:** 1Department of Anesthesiology and Pain Medicine, Soonchunhyang University Hospital Cheonan, 31, Suncheonhyang 6-gil, Dongam-gu, Cheonan 31151, Korea; anesth70@schmc.ac.kr (J.H.C.); nskim1977@schmc.ac.kr (N.S.K.); c75501@schmc.ac.kr (Y.H.S.); dyflam@schmc.ac.kr (H.S.J.); blau00@schmc.ac.kr (H.R.C.); susu1987@naver.com (J.S.C.); 135730@schmc.ac.kr (H.C.J.); 118541@schmc.ac.kr (J.S.P.); 2Department of Neurosurgery, Soonchunhyang University Hospital Cheonan, 31, Suncheonhyang 6-gil, Dongam-gu, Cheonan 31151, Korea; jmstarry21@naver.com; 3Department of Orthopaedic Surgery, Soonchunhyang University Hospital Cheonan, 31, Suncheonhyang 6-gil, Dongam-gu, Cheonan 31151, Korea; kwj9383@hanmail.net; 4Department of Radiology, Soonchunhyang University Hospital Cheonan, 31, Suncheonhyang 6-gil, Dongam-gu, Cheonan 31151, Korea; 98480@schmc.ac.kr

**Keywords:** young, spinal anesthesia, hematoma, spinal, magnetic resonance imaging

## Abstract

Spinal intradural hematoma (SIH) is a rare condition which can cause neurological sequelae such as permanent motor weakness and sensory loss in the lower extremities. Herein, we describe a case of SIH following spinal anesthesia. The patient was a 30-year-old man who underwent treatment for accessory navicular syndrome at our department. The patient was not receiving anticoagulation therapy, and spinal anesthesia was thus selected. No symptoms of hematoma were observed in the immediate postoperative period, but the patient complained of pain in both buttocks on postoperative day 5. However, neither motor weakness nor sensory loss were observed. Additionally, as the radiating pain extending to the lower extremities typical of neurological pain was not observed, musculoskeletal pain was suspected. Magnetic resonance imaging revealed intradural hematomas at L4-5 and S1. Conservative treatment and follow-up evaluations were performed to ensure that additional neurological sequelae did not occur. Six months after symptom onset, his pain Numeric Rating Scale score was 0, and no other neurological findings were observed. However, in patients who undergo spinal anesthesia, localized pain in the back without other neurological symptoms and lack of radiating pain may be associated with more than musculoskeletal pain. Such patients must be continuously monitored.

## 1. Introduction

Spinal intradural (subdural and subarachnoid) hematoma (SIH) is caused by lumbar spinal punctures, underlying neoplasm, old age, infection, spinal surgery, anticoagulation therapy, and bleeding disorders (hemophilia, leukemia, thrombocytopenia) [1,2,3,4]. Neuraxial hematoma after spinal anesthesia, which is commonly performed for lower extremity anesthesia, is the rarest cause, with an incidence of <1%. Epidural hematoma is the most commonly observed (75%) cause, and the incidences of subarachnoid hematoma and subdural hematoma are 15.7% and 4.1%, respectively [5,6]. Although SIH is rare, it can cause neurological sequelae such as permanent motor weakness and sensory loss in the lower extremities. Such complications must be carefully monitored after spinal anesthesia [7]. Herein, we describe the case of a 30-year-old patient with no specific medical history who developed SIH following spinal anesthesia. We report the details of the case and review the possible causes of SIH and adequate countermeasures.

## 2. Case Presentation

This study was approved by the Institutional Review Board (IRB No. 2022-02-009), and consent was obtained from the patient for the publication of this report.

This case was a 30-year-old male patient who works as a motorcycle deliveryman. His body mass index in the historical record was 26.7, and he had no specific medical history or drug use, except that he had stage 1 obesity. He injured himself by falling while walking on the street, and the subsequent pain persisted to the extent that it interfered with walking, with pain on the inside of the foot, and the patient visited the hospital. On radiological findings, an accessory bone was found in the navicular bone (Figure 1), so the patient decided to undergo a modified Kidner procedure. A blood test performed 1 day preoperatively revealed a platelet count of 256,000/µL, a prothrombin time (international normalized ratio) of 1.10, and an activated partial thromboplastin clotting time of 35.9 s. The patient had not received anticoagulation therapy for 6 months prior to the operation, and thus, spinal anesthesia was selected. After the patient was placed in the lateral position, a 24-gauge Quincke needle was inserted into the midline between the fifth lumbar vertebra and first sacral vertebra. The first insertion failed, as the bone was touched. On the second attempt, penetration of the dura mater was detected by a pop sign. As cerebrospinal fluid (CSF) was withdrawn, 10 mg of bupivacaine was administered. Blood was not observed through the needle in either attempt. Upon physical examination, a sensory block was confirmed at thoracic dermatome level 6, and the operation commenced. The patient exhibited no specific symptoms until he was discharged from the recovery room. Immediately after the operation, the patient complained of pain in the surgical site, which was somewhat relieved after intravenous injections of Nefopam 60 mg and Propacetamol 1 g per day for two days. Dressing at the surgical site was performed daily, and antibiotics, Flomoxef sodium 500 mg, was administered. There were no symptoms or signs of infection at the surgical site. Two days after surgery, the patient was able to walk using crutches. On the fifth day postoperatively, the patient complained of burning and sharp pain in both buttocks, which worsened at night. Physical examination revealed motor grade 5 for both legs, which did not indicate motor weakness. The patellar and Achilles tendon reflexes were normal, and neurological abnormalities such as urinary retention and sensory loss in the lower extremities were not observed. Additionally, as the radiating pain extending to the lower extremities typical of neurological pain was not observed, musculoskeletal pain was suspected. Thus, the nonsteroidal anti-inflammatory drug (NSAID), Celebrex, 200 mg, and muscle relaxant, Eperisone, 50 mg, were each administered twice a day, and extracorporeal shock wave therapy (ESWT, 0.23–0.45 mJ/mm^2^, 1500 to 2000 times/region, 7 Hz) was provided; however, the symptoms were only temporarily improved and later worsened again. The patient complained of pain in the bilateral sacrum, and a Patrick’s test was conducted. Although the result was negative, a joint injection was conducted on both sides to evaluate inflammation of the sacroiliac joint [8]. However, the patient’s symptoms did not improve. Although there are no clear neurological symptoms, since symptoms appear after spinal anesthesia, it was necessary to reliably exclude spinal intradural hematoma. Thus, magnetic resonance imaging (MRI) was conducted to allow accurate diagnosis. Non-contrast MR images revealed intradural hematomas at L4-5 and S1. T1 images showed hyperintensity and T2 images showed hypointensity, suggesting early subacute hematomas. It was assumed that the hematomas developed after spinal anesthesia (Figure 2 and Figure 3).

After MRI, a conventional conservative treatment and continuous physical examination were provided to ensure that additional neurological sequelae did not occur. As no serious complications such as motor weakness, sensory loss, urinary distension, or any neurological sequelae were observed and the pain was relieved, it was decided to monitor the patient’s progress with follow-ups periodically after discharge. Three months after discharge, contrast-enhanced MR images were obtained to determine the cause of the hematoma and assess the improvement of the condition. MRI revealed that the intradural hematoma at L4-5 had decreased in size and that the nerves were displaced to the left side. There were no findings of tumor or vascular malformation (Figure 4).

Six months after symptom onset, the pain Numeric Rating Scale (NRS) score was 0, and no other neurological findings were observed (Figure 5, flow diagram).

## 3. Discussion

Although the patient showed findings indicative of SIH and pain in the bilateral sacrum, paralysis of the lower extremities or serious neurological findings such as sphincter dysfunction were not observed. His symptoms improved over time. In patients who develop SIH after spinal anesthesia, the symptoms generally manifest 3–4 days later [7]. In the current patient, no symptoms were immediately observed after spinal anesthesia, and buttock pain was first observed five days postoperatively. As tumors and vascular malformation are possible causes of SIH [9], contrast-enhanced spinal MRI was conducted three months later. However, no specific findings were observed.

In spinal hematoma, although blood does not press on the nerves, it may have neurotoxic effects on the spinal cord [3], causing pain. Once blood clots have formed, the spinal cord may be damaged, and arachnoid fibrosis may occur [2]. Therefore, persistent complaints of pain after spinal anesthesia must be diagnosed and treated promptly to identify possible hematoma.

SIH is caused by various factors; however, the pathogenesis of hematoma has not been clearly established [10,11]. Additionally, even if there is bleeding in the subarachnoid space, it is highly likely that the blood is diluted by the flow of CSF, causing it to disappear. Therefore, hematoma is rarely observed in the subarachnoid space, and the case reported here is thus quite rare [12].

In other reported cases of SIH at the lumbar level (), hematomas were observed after spinal anesthesia in patients without any relevant medical history, as in our case [6,13]. Moreover, in other patients, the hematoma was caused by coagulopathy [3,14,15].

In one unique case, it was reported that bleeding of a cranial hematoma may have descended toward the spine following gravity, and that this may have caused a spinal hematoma. Therefore, it was recommended that brain imaging also be conducted if the patient complained of a headache [16] As in the cases shown in Table 1, hematoma is commonly diagnosed by physical examination or spinal MRI after observing neurological symptoms suggestive of spinal abnormalities. However, as in the current patient who exhibited no signs of coagulopathy, complained of localized pain in the sacrum or lower back after spinal anesthesia, and did not exhibit neurological symptoms, hematoma may be easily overlooked.

A total of 33 previous cases of spinal hematoma have been reported between 1911 and 1981. Eleven patients had underlying diseases and died, and nine patients achieved complete neurological recovery, indicating that the prognosis of spinal hematoma is generally poor [7]. However, in several recent cases [1,3,17], early diagnosis following the improvement of diagnostic and surgical techniques led to complete recovery without sequelae (Table 1).

In cases in which it was difficult to determine the definite cause, it is possible that the rapid increase in abdominal pressure of the large radiculomedullary vessels and abundant capillary network in the subarachnoid space may rupture and form a hematoma, spreading to the subdural space [10,18]. Other causes include bleeding from aggressive needling or technical issues related to the piercing of the anterior portion of the subarachnoid space.

In patients who exhibit hematoma-induced neuropathy symptoms and do not undergo surgery, there is a 63.6% possibility of developing serious sequelae such as permanent paralysis of the lower extremities and sphincter dysfunction. However, in those who undergo emergency surgery immediately after symptom onset, the risk of serious sequelae is significantly lower, at 17.9% [2]. Therefore, if serious neurological damage is suspected, prompt surgical treatment is required.

However, when a patient who is taking anticoagulants or suffering from coagulopathy develops hematoma due to spinal anesthesia, and a decompression surgery is urgently performed on the spine, difficulties in bleeding control due to surgery and intraoperative bleeding may further compress the spinal cord, leading to additional damage [19]. Therefore, for patients with coagulopathy, general anesthesia is preferred over spinal anesthesia.

When there are no serious neurological complications caused by SIH, conservative treatment is commonly provided. Provided complications such as motor weakness are absent, conservative treatment is associated with a good prognosis of pain symptom recovery within a few hours at the earliest [19]. In the current patient, MRI demonstrated that the hematoma was not partially confined to the intradural space but was spread widely from L4 to S3, and that there were risks of additional neurological complications. However, in the early period after hematoma was identified, no symptoms other than pain were observed. The symptoms were relieved after three months of conservative treatment. At six months postoperatively, the pain NRS score was 0, and no other neurological sequelae were observed. 

## 4. Conclusions

In this report, a young patient with no underlying disease who was not receiving anticoagulant therapy developed SIH after spinal anesthesia. Although patients may not be considered to be at high risk of spinal hematoma, the possibility of a hematoma must be carefully monitored after surgery. Furthermore, in patients who undergo spinal anesthesia, localized pain in the back without other neurological symptoms and lack of radiating pain from the lumbar level may be associated with more than simple musculoskeletal pain. Such patients must be continuously monitored for potentially lethal or permanent neurological complications and undergo MRI for an accurate diagnosis.

## Figures and Tables

**Figure 1 ijerph-19-04845-f001:**
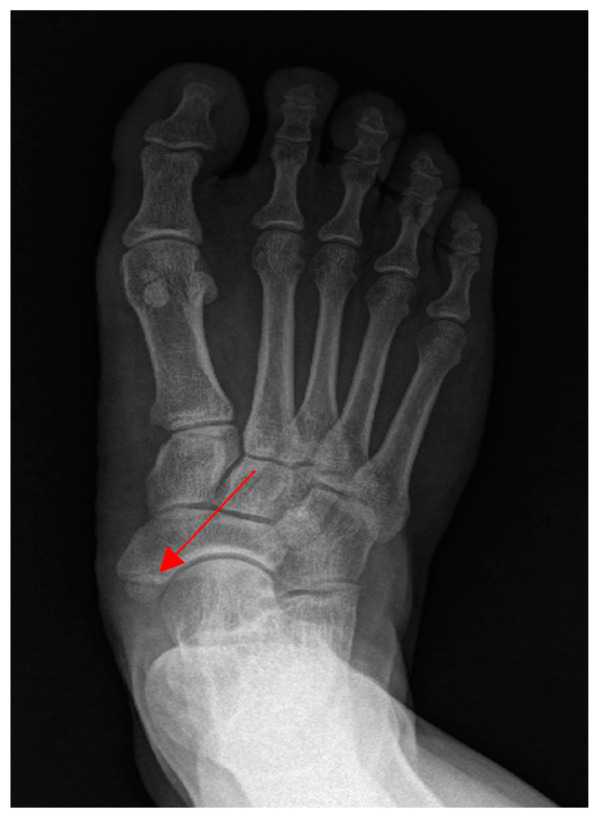
Patient’s accessory navicular bone (Arrow).

**Figure 2 ijerph-19-04845-f002:**
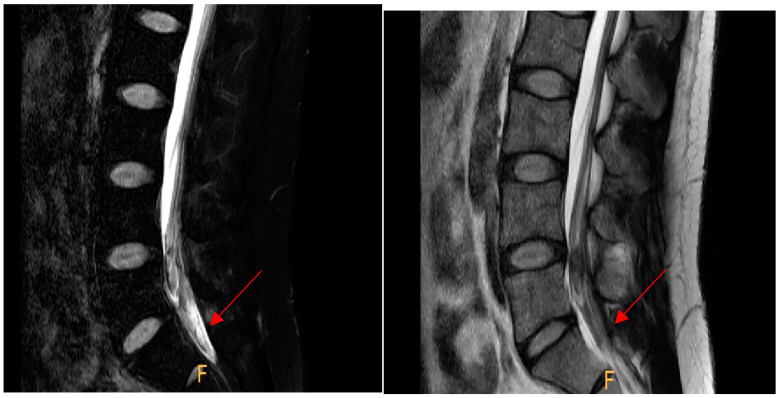
Hematoma (arrow) with a hyperintense signal at L4-5 and S1 on sagittal magnetic resonance imaging (MRI) (T1) on the **left**, and hematoma (arrow) with a hypointense signal on sagittal MRI (T2) on the **right**.

**Figure 3 ijerph-19-04845-f003:**
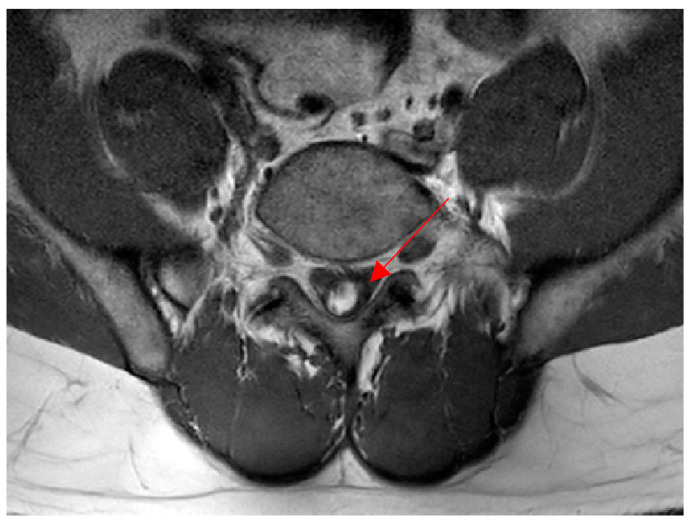
Hematoma (arrow) on the intradural side on magnetic resonance imaging (T2). In the horizontal section, the hematoma was observed on the spinal intradural side.

**Figure 4 ijerph-19-04845-f004:**
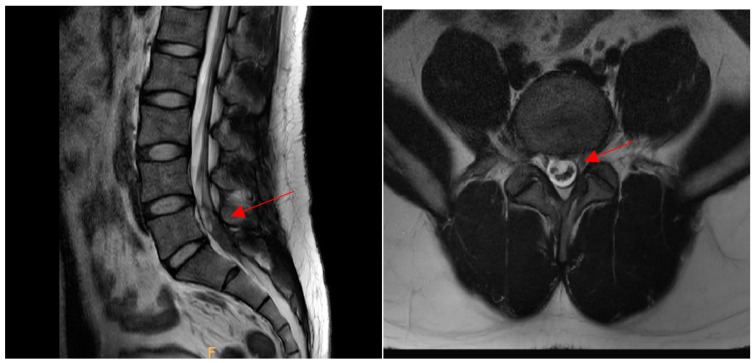
Magnetic resonance imaging (T2) showed a slight decrease in the size of the hematoma on the **left** side (arrow). The intradural hematoma on the **right** side was also decreased in size (arrow).

**Figure 5 ijerph-19-04845-f005:**
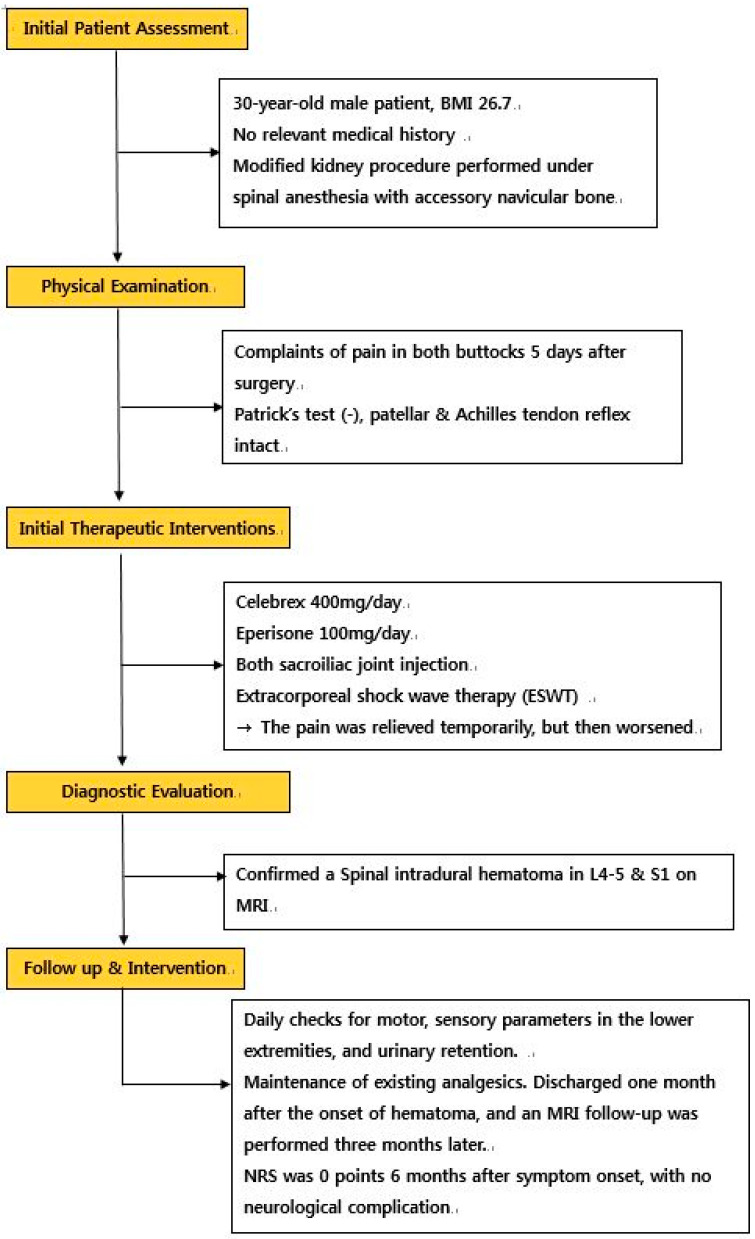
Flow diagram.

**Table 1 ijerph-19-04845-t001:** Spinal intradural (subarachnoid or subdural) hematoma cases including the lumbar level.

Case Study	Age (Years)	Cause	Hematoma Type	Symptoms	Treatment	Outcome
Boukobza et al. (2001) [14]	54	Antivitamin K treatment Spontaneous	Subdural (T9-L1)	Lower back pain, saddle pain, urinary retention	Laminectomy (T10-L4)	Complete recovery
Mashiko et al. (2006) [20]	18	Trauma	Subdural (L5-S2)	Headache, lower back pain	Observation	Complete recovery
Lam et al. (2008) [7]	Elderly	Spinal anesthesia	Subarachnoid (L4)	Lower back pain, buttock pain, right leg pain	Conservative treatment	Complete recovery
Koyama et al. (2009) [15]	39	HELLP (Hemolysis, Elevated Liver enzymes, and Low platelet count) syndrome, Spinal anesthesia	Subarachnoid (L2-S1)	Numbness of the thigh and toes, urinary retention	Conservative treatment	Complete recovery
Moon et al. (2013) [16]	39	Chronic cranial subdural hematoma	Subdural (L4-S2)	Headache, lower back pain, pain radiating down both legs	Evacuation of cranial hematoma	Complete recovery
Bruce–Brand et al. (2013) [3]	76	Warfarin treatment Spontaneous	Intradural (L1-L4)	Flaccid paraparesis in both legs, sensory weakness below T12	Laminectomy (T12-L4), intradural hematoma evacuation	Remaining motor weakness (ASIA grade C)
Jeon et al. (2013) [6]	33	Unknown Spinal anesthesia	Subarachnoid (L5)	Lower back pain, numbness of the lower limbs	Observation	Complete recovery
Basaran et al. (2014) [17]	27	Lumboperitoneal shunt operation	Intradural (L2-3)	Urinary and fecal incontinence with paraparesis	L2-L3 total laminectomy	Permanent paraparesis
Avecillas-Chasin et al. (2015) [1]	79	Spinal anesthesia	Intradural (L1-L3)	Motor and sensory weakness	Laminectomy (L1-L3), intradural hematoma evacuation	Unable to walk without assistance
Cui et al. (2015) [13]	45	Unknown	Subdural (L4-S3)	Saddle pain and dysuria	Hematoma evacuation	Complete recovery
Jang et al. (2017) [5]	59	Unknown	Subarachnoid (L3-L4)	Headache, nausea, vomiting, neck stiffness, lower back pain	Laminectomy (L3-4), hematoma evacuation	Complete recovery

## Data Availability

Data sharing is not applicable to this article as no datasets were generated or analyzed during the current study.

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
