# Peer review of "Spinal Intradural Hematoma after Spinal Anesthesia in a Young Male Patient: Case Report and Review of the Literature"

_ijerph, 2022, doi:10.3390/ijerph19084845_

Round 1

Reviewer 1 Report

Thank you for submitting your manuscript “Spinal intradural hematoma after spinal anesthesia in a young male patient: Case report and review of the literature.” Adverse events should be reported after interventions as they are often under-reported. Reporting of such effects is needed to improve health care by increasing clinical knowledge and awareness. A few comments are detailed below.

Overall, the language mixes general health (e.g. knee reflex instead of patella, both legs not bilaterally) and discipline-specific language (e.g. Kidner procedure). The audience the text is being pitched to is unclear. 

A sentence stating the importance of reporting such a case study is missing in the background. If the purpose is educational material, consistent medical language should be used, re the previous point.

The case presentation could be strengthened by splitting into more paragraphs (e.g. clinical history, procedure/intervention, post-surgical management). This would also aid in separating the indicated surgery for the navicular and the anaesthesia procedure.

 I recommend including more clinical history details (e.g. age, medication history, comorbidities, any other clinically significant information).

Some sentences are generic and could benefit from being more specific details e.g.

  • Line 51, depending on the audience, a definition or clinical presentation of the navicular syndrome requiring it be a surgical intervention would be beneficial.
  • Line 65 “both sacral areas”. Do the authors mean the sacro-iliac joints bilaterally?
  • Line 69, what is meant by “musculoskeletal pain”? Do the authors mean that the lower limb had non-dermatomal pain? 
  • Line 70-1, what was the dose of the medicines and shock wave therapy? Was it in therapeutic ranges, and the patient compliant in taking the therapies?
  • Line 89, what was the conservative treatment?
  • Line 92, where was the patient discharged? The ED, hospital, tertiary care?

The discussion includes a summary of relevant articles. How this was derived is unclear. It is usual to resent such information in this format. It feels like it is trying to emulate a systematic review without doing one. 

Author Response

Thank you for submitting your manuscript “Spinal intradural hematoma after spinal anesthesia in a young male patient: Case report and review of the literature.” Adverse events should be reported after interventions as they are often under-reported. Reporting of such effects is needed to improve health care by increasing clinical knowledge and awareness. A few comments are detailed below.

Overall, the language mixes general health (e.g. knee reflex instead of patella, both legs not bilaterally) and discipline-specific language (e.g. Kidner procedure). The audience the text is being pitched to is unclear. 

 Answer) We have changed this to ‘patellar and Achilles tendon reflexes’.

A sentence stating the importance of reporting such a case study is missing in the background. If the purpose is educational material, consistent medical language should be used, re the previous point.

 Answer) We have rechecked the medical terminology.

The case presentation could be strengthened by splitting into more paragraphs (e.g. clinical history, procedure/intervention, post-surgical management). This would also aid in separating the indicated surgery for the navicular and the anaesthesia procedure.

 Answer) The reason why the patient underwent surgery and the imaging findings were described, and analgesics for post-operative pain and antibiotics after surgery were additionally described.

 I recommend including more clinical history details (e.g. age, medication history, comorbidities, any other clinically significant information).

 Answer) The 30-year-old male patient had no specific medical history and was not taking any medications.

Some sentences are generic and could benefit from being more specific details e.g.

Line 51, depending on the audience, a definition or clinical presentation of the navicular syndrome requiring it be a surgical intervention would be beneficial.

Answer) Navicular syndrome was additionally described in the texts, with relevant photos.

Line 65 “both sacral areas”. Do the authors mean the sacroiliac joints bilaterally?

Answer) Pain in the sacroiliac joint occurs in both buttocks. The term sacral area was ambiguous, therefore we changed it to Buttock in the texts.

Line 69, what is meant by “musculoskeletal pain”? Do the authors mean that the lower limb had non dermatomal pain? 

Answer) The pain was limited to the buttock area, and there was no radiating pain or motor or sensory weakness traveling down to the legs. Thus, we described it as musculoskeletal pain considering it was pain related to muscle and bone, not nerve origin pain.

Line 70-1, what was the dose of the medicines and shock wave therapy? Was it in therapeutic ranges, and the patient compliant in taking the therapies?

Answer) Celebrax 200mg and Eperison 50mg were taken twice a day, and ESWT (Extracorporeal shock wave therapy (0.23-0.45 mJ/mm2, 1500 to 2000 times/region, 7 Hz) was administered for two days. However, This had only a temporary effect, and the patient complained of persistent pain. We added this information to the texts. In general, it is recommended to take Celebrax twice a day, totaling 400 mg per day, with Eperison 50 to 150 mg per day. ESWT was performed by setting the value used for general muscle pain.

Line 89, what was the conservative treatment?

Answer) Drug treatment, physical therapy, and injection treatment, excluding surgical treatment, are conservative treatments.

Line 92, where was the patient discharged? The ED, hospital, tertiary care?

 Answer) The patient was observed for one month after being diagnosed with a spinal hematoma, during which no neurological sequelae were seen. After a month, the patient was discharged home and asked to visit the hospital immediately when neurological sequelae occur. There were periodic visits to the hospital for follow-ups. We have added this information in the texts.

The discussion includes a summary of relevant articles. How this was derived is unclear. It is usual to resent such information in this format. It feels like it is trying to emulate a systematic review without doing one. 

Answer) We reviewed the relevant evidence because it was necessary to compare the symptoms, treatment process, and prognosis after hematoma with other cases of spinal intradural hematoma that occurred at the same lumbar level as the case in this study. We reviewed case studies and literature investigating hematoma cases.

Reviewer 2 Report

Dear authors

this paper is correct, but it needs some improvments and sharpness

First of all, describe better the papients status (which work he does, its Body mass index).

Please describes the flow chart or selected article (which mesh) did you use a prisma checklist? something else?

Please describe better somothing about needles? why don't you use 25 or 27 Whitacre needle?

I'm favorable to case report: anyway many jorunals want and reccomend an high level and quality of presentation, so I suggest to you a take home message figure, especially designed as algorithm.

Please, increase the level of interest of the paper for readers and stakeholders.

Author Response

Dear authors

this paper is correct, but it needs some improvments and sharpness

First of all, describe better the papients status (which work he does, its Body mass index).

Answer) The patient was a motorcycle delivery man, and his Body mass index was 26.7. We have added this information to the text.

Please describes the flow chart or selected article (which mesh) did you use a prisma checklist? something else?

Answer) In the last item, the patient's case progression was organized in a flow chart.

Please describe better somothing about needles? why don't you use 25 or 27 Whitacre needle?

Answer) It would have been better to use the Whitacre needle to have a lower incidence of post-dural puncture headache, but we did not have it in stock. Although the 25 Gauge or 27 Gauge needle has a lower incidence of post-dural puncture headache, the 24 Gauge was used because it could confirm the CSF flow more reliably.

I'm favorable to case report: anyway many jorunals want and reccomend an high level and quality of presentation, so I suggest to you a take home message figure, especially designed as algorithm.

Answer) In the last item, the patient's case progression was organized in a flow chart.

Please, increase the level of interest of the paper for readers and stakeholders.

Reviewer 3 Report

This is a very well written case report article, however there are some points that could strengthen the evidence presented by the authors and further clarify the messages for the readership of the journal:
1. p.4, lines 119-20: Please revise the phrase "Additionally ... the flow of CSF". The reader might assume that bleeding is caused by CSF flow.
2. p.5, line 136: Please explain why the readership should be specifically interested in the number of cases reported between 1911 and 1981 (and not in the total number of reported cases) and how this time interval was chosen.
3. Spinal intradural hematoma after spinal anesthesia, although rare, has been reported previously and it still is the worst case scenario feared by anesthiologists in this type of anesthesia. What makes this case report mentionable is perhaps the absence of neurological deficits. Therefore, this perhaps should be reflected somehow in the title.

Author Response

This is a very well written case report article, however there are some points that could strengthen the evidence presented by the authors and further clarify the messages for the readership of the journal:

  1. p.4, lines 119-20: Please revise the phrase "Additionally ... the flow of CSF". The reader might assume that bleeding is caused by CSF flow.

Answer) We have changed the text to 'Even if there is bleeding in the subarachnoid space, it is highly likely that the blood will be diluted by the flow of CSF, causing it to disappear.'

  1. p.5, line 136: Please explain why the readership should be specifically interested in the number of cases reported between 1911 and 1981 (and not in the total number of reported cases) and how this time interval was chosen.

Answer) Spinal intradural hematoma can sometimes be fatal; however, the prognosis is significantly improved compared to the past due to the development of diagnosis and treatment.

  1. Spinal intradural hematoma after spinal anesthesia, although rare, has been reported previously and it still is the worst case scenario feared by anesthiologists in this type of anesthesia. What makes this case report mentionable is perhaps the absence of neurological deficits. Therefore, this perhaps should be reflected somehow in the title.

Answer) Thank you for your useful and relevant comments. It would have been nice to add that there was no mention of neurological defects in the title; however, it is difficult to change this because the title of the case report has been approved by the IRB(Instituional Review Board), and the title submitted to the journal cannot be different.

Round 2

Reviewer 2 Report

Dear authors

The paper has been improved now it is ready for publication